# Expression of Toll-like receptors in oral squamous cell carcinoma

**Peter Rusanen**[ID][1][☙]*, **Emilia Marttila**[2][☙], **Sajeen Bahadur Amatya**[3], **Jaana Hagström**[1], **Johanna Uittamo**[2,4], **Justus Reunanen**[3], **Riina Rautemaa-Richardson**[ID][5,6][☙], **Tuula Salo**[2,7,8][☙]

**1** Department of Bacteriology and Immunology, Haartman institute, University of Helsinki, Helsinki, Finland, **2** Department of Oral and Maxillofacial Surgery, Helsinki University Hospital and University of Helsinki, Helsinki, Finland, **3** Biocenter Oulu & Cancer and Translational Medicine Research Unit, University of Oulu, Oulu, Finland, **4** Research Unit on Acetaldehyde and Cancer, University of Helsinki, Helsinki, Finland, **5** Division of Infection, Immunity and Respiratory Medicine, School of Biological Sciences, NIHR Manchester Biomedical Research Centre (BRC) at the Manchester Academic Health Science Centre, The University of Manchester, Manchester, United Kingdom, **6** Manchester University NHS Foundation Trust, Wythenshawe Hospital, Manchester, United Kingdom, **7** HUSLAB, Department of Pathology, Helsinki University Central Hospital, University of Helsinki, Helsinki, Finland, **8** Department of Diagnostics and Oral Medicine, Institute of Dentistry, University of Oulu, Oulu, Finland

☙ These authors contributed equally to this work.
* peter.rusanen@helsinki.fi

**Data Availability Statement:** All relevant data are within the manuscript and its Supporting Information files.

**Funding:** This work was supported by the Suomen Hammaslääkäriseura Apollonia (Finnish Dental

## Abstract

Almost 380,000 new cases of oral cancer were reported worldwide in 2020. Oral squamous cell carcinoma (OSCC) accounts for 90% of all types of oral cancers. Emerging studies have shown association of Toll-like receptors (TLRs) in carcinogenesis. The present study aimed to investigate the expression levels and tissue localization of TRL1 to TRL10 and NF-κB between OSCC and healthy oral mucosa, as well as effect of Candida colonization in TRL expression in OSCC. Full thickness biopsies and microbial samples from 30 newly diagnosed primary OSCC patients and 26 health controls were collected. The expression of TLR1 to TLR10 and NF-κB was analyzed by immunohistochemistry. Microbial samples were collected from oral mucosa to detect *Candida*. OSCC epithelium showed lower staining intensity of TRL1, TRL2 TRL5, and TRL8 as compared to healthy controls. Similarly, staining intensity of TRL3, TRL4, TRL7, and TRL8 were significantly decreased in basement membrane (BM) zone. Likewise, OSCC endothelium showed lower staining intensity of TLR4, TLR7 and TLR8. Expression of NF-κB was significantly stronger in normal healthy tissue compared to OSCC sample. Positive correlation was found between the expression of NF-κB, TRL9 and TRL10 in basal layer of the infiltrative zone OSCC samples (P = 0.04 and P = 0.002, respectively). Significant increase in TRL4 was seen in BM zone of sample colonized with Candida (P = 0.01). According to the limited number of samples, our data indicates downregulation of TLRs and NF-κB in OSCC, and upregulation of TLR4 expression with presence of Candida.

Society Apollonia) and the Helsingin ja Uudenmaan Sairaanhoitopiiri (Helsinki University Central Hospital) Research Funds. JR was supported by Academy of Finland. RR-R was supported by NIHR Manchester Biomedical Research Centre. The funders had no role in study design, data collection, and analysis, decision to publish, or preparation of the manuscript.

**Competing interests:** The authors have declared that no competing interests exist.

## Introduction

Oral cavity cancer is ranked as one of the most common malignancies in the world with its prevalence rising yearly. Europe diagnoses more than 100,000 new cases each year. Over 90% of all oral malignancies are attributed to oral squamous cell carcinomas (OSSCs), with the mobile tongue being the most affected area [1]. Regardless of advances in treatment modalities, no significant improvement has been observed in the 5-year survival rate of OSCC in decades [2]. Smoking and drinking are deemed to be the key risk factors of OSCCs [3, 4]. Chronic candidiasis has been linked to the pathogenesis of oral carcinoma; however, the mechanism is unknown [5]. Several studies have confirmed that *Candida* species are able to produce mutagenic levels of carcinogenic acetaldehyde *in vitro* [6–8].

The role of chronic inflammation and the innate immune system in the development of cancer is widely recognized and a strong link between chronic inflammation and many types of cancers have been reported [9–12]. Anti-apoptotic effects of nuclear factor-kappaB (NF-kappaB), induction of tissue repair response and oxidative DNA damage are among the various mechanisms of inflammatory response that aid in promotion of carcinogenesis [9, 13, 14].

Toll like receptors (TLRs) are a family of trans-membranous pattern recognition receptors that recognize molecular patterns of microbial pathogens as well as those of endogenous origin. They have a key role in the activation of the innate immunity and provide a quick and highly efficient response to pathogens and host-derived molecules. TLRs are involved in maintenance of various aspects of tissue homeostasis via regulation of inflammatory and tissue repair responses [15]. Ten different types of TLRs are recognized in humans, TLR1-TLR10. Along with the cells of the immune system, these TLRs are also expressed by various non-immune cells such as basement membrane cells of oral mucosa and keratinocytes of skin [16, 17]. Additionally, body fluids such as saliva, plasma, breast milk, pleural fluid and amniotic fluid have been found to harbor soluble forms of TLRs [18–21]. Endothelial cells have been shown to express one or more TLRs when triggered by a stimulus such as infection by pathogens or by tissue damage signaling molecules [22]. The expression and signaling of TLRs in endothelial cells is a key factor to regulate angiogenesis. While angiogenic process is deemed crucial for tissue repair, endothelial dysfunction promoted by TLRs contributes to tissue damage in inflammatory and autoimmune diseases including rheumatoid arthritis, systemic lupus erythematosus, atherosclerosis, and even cancer [22, 23]. Indeed, accumulating evidence points to a crucial role of TLRs in angiogenesis which is required for the growth of some tumors [23]. To date, the understanding of the ligand recognition, signaling and biological functions of TLR1-TLR9 is fairly clear. In contrast, the agonists, signaling, and function of TLR10 remains largely unknown [24].

Stimulation of various TLRs leads to the activation of different transcription factors, such as nuclear factor-kappaB (NF-κB). NF-κB regulates the genes that encode for expression of cytokines, chemokines, and co-stimulatory molecules such as TNF-α, IL-1β, and leukocyte and vascular adhesion molecules [25]. These immune and pro-inflammatory mediators play essential roles in recruiting various inflammatory cells into the infection sites and activating the adaptive immune response later in infection. Some of these pro-inflammatory mediators can also activate NF-κB, and a type of positive regulatory loop may be formed that would aggravate and perpetuate the local inflammatory reactions. Since NF-κB plays a significant role in innate immunity, adaptive immunity, and cell proliferation processes, its activity is tightly regulated. Dysregulation of NF-κB activation pathway at any stage can lead to chronic inflammation, autoimmunity, and cancer [25, 26].

TLRs regulate a wide range of biological responses including immune and inflammatory responses during carcinogenesis [12]. TLRs may promote carcinogenesis through tumor

promoting inflammatory signals, anti-apoptotic pathways, cell proliferative and fibroblast activation mechanisms, influencing either tumor cells or the tumor microenvironment [27]. Yet, the association of TLRs in OSCC is conflicting due to lack of adequate data and discrepancies between published studies [28]. However, several studies have demonstrated a role of TLR4 in the pathogenesis of cancer.

Downregulation of TLR4 has been reported to inhibit tumour growth and inflammation, cytokine secretion and to suppress metastasis of carcinoma in several cancers [29–32]. Moreover, according to the meta-analysis of Hao et al. 2018, the elevated expression of TLR4 in cancer patients is associated with poor overall survival and shorter disease-free survival [33]. In addition, a correlation between high TLR4 expression and worse survival rate in OSCC patients has been reported [34] and TLR-2, -4, and -9 seemed to predict invasive tumor growth [35]. Also, overexpression of TLR3 was associated with poorly differentiated OSCC and the polymorphism of TLR3 has been associated with the prognosis of transformation to malignant lesions in the mouth [36, 37].

The primary aim of this study was to investigate the expression levels and tissue localization of TLR1-TLR10 and NF-κB in mucosal biopsies from oral squamous cell carcinoma in comparison to that of healthy oral mucosa. In addition, we analyzed whether the presence of *Candida* colonization affects TLR expression in oral squamous cell carcinoma.

## Material and methods

### Study subjects

A total of 56 voluntary patients, 30 with newly diagnosed primary oral squamous cell carcinoma (OSCC) and 26 healthy controls (HC) treated at the Department of Oral and Maxillofacial Surgery, Helsinki University Hospital during 2007–2011 were enrolled in this study (Table 1). Patients who had received antimicrobial therapy (i.e. antibiotics, antifungals, or antiviral agents) within the past seven days and those diagnosed with HIV or hepatitis virus infection were excluded. All study patients signed an informed consent before inclusion. The study has been approved by the Ethics Committee of the Helsinki University Central Hospital (Ethical approval number 126/E6/07 25.4.2007). Written consent from all the subjects were taken according to the declaration of Helsinki. Of the 30 healthy controls (HC) enrolled in the study, 26 were included in the final analyses as four had inadequate amount of tissue for histopathological analyses.

**Table 1. Patient demographics.**

|  | OSCC | HC |
|---|---|---|
| **Total number** | 30 | 26 |
| Female: male | 12:18 | 17:9 |
| **Age in years (range)** | 64 (31–85) | 30.8 (18–56) |
| **Location of the lesion** |  |  |
| Tongue | 9 |  |
| Mandibular gingiva | 6 |  |
| Maxilla | 6 |  |
| Floor of the mouth | 5 |  |
| Palate | 2 |  |
| Alveolar ridge | 2 |  |

## Collection of histopathological samples

Full thickness biopsies including epithelial and stromal tissue were collected from OSCC patients from the site of active disease process according to normal clinical procedures. The biopsies from healthy control patients were collected from the buccal mucosa at the incision site immediately after surgical extraction of a retained wisdom tooth. The samples were fixed in 10% buffered formalin and embedded in paraffin.

## Collection of microbial samples

Microbial samples for detection of *Candida* were collected using the filter paper sampling method as described in Rusanen et al., 2009 [38]. The samples were taken from the oral mucosa using a hydrophilic mixed cellulose ester MF-Millipore membrane filter (GSWP01300; Millipore Inc., Billerica, MA, USA, pore size 0.22 μm, diameter of 13 mm) [38]. The filter paper was placed gently on the oral mucosa for 30 s with the glossy side placed against the mucosa, after which it was placed into a sterile test tube containing 5 mL of sterile saline solution. All samples were cultured within 1 h of collection. Before culture, the samples were agitated for 30 s with five sterile Ø 3 mM glass beads. For the detection of yeasts, the samples were diluted 10-fold and 100 μl of the dilution, and the neat suspension were cultured on Sabouraud Dextrose Agar (SP; Saboraud Dextrose Agar [Lab M], Bacto Agar [Difco Laboratories, Basel, Switzerland] supplemented with penicillin [100,000 iu/mL] and streptomycin) and incubated for 48 hours at 37˚C. After incubation, the numbers of yeasts were enumerated as colony forming units (CFU).

## Immunohistochemical staining

Tissue sections, 4 μm in thickness, were prepared from the paraffin embedded samples and applied to glass slides. The sections were deparaffined in xylenes, followed by rehydration in graded ethanol, and washed in deonized $H_2O$. To expose the antigenic determinants after formalin fixation and paraffin embedding, the sections were incubated in pepsin for 30 min at room temperature. Endogenous peroxidase activity was quenched in the sections by incubating in hydrogen peroxidase in methanol.

The optimal primary antibody immunoglobulin G (IgG) concentration for immunohistochemistry was selected for each TLR antibody in pilot experiments and according to our previous study [17, 39]. All tissue samples were stained using this antibody concentration to allow comparisons between staining intensity. The final IgG concentrations of the polyclonal anti-human antibodies used in this study are shown in the Table 2. The TLRs were visualized as specified in user manual (catalogue nos., PK-4001 and PK-4005; Vectastain ABC kit; Vector Laboratories, Peterborough, England).

For the immunohistochemical staining with NF-κB, the tissue sections were buffered in citrate, pH 6.0 and heated 10 minutes in microwave oven and incubated for one hour in room temperature with an optimally diluted NF-κB antibody according to our previous study [17] (Table 2). After the primary antibody incubation, the tissue sections were incubated separately with Dako REAL™ EnVision™ kit using Dako automated immunostaining instruments. The reactions were visualized by Dako REAL™ DAB+ Chromogen according to manufacturer instruction (catalogue number K5007, Dako Glostrup Denmark).

Tissue samples from chronic periodontitis were used as positive controls [39, 40]. Negative controls were obtained by omission of the primary antibodies. All the specimens were stained with periodic acid-Schiff (PAS) method to determine the presence or absence of secondary candidiasis.

**Table 2. The optimal IgG concentrations of the polyclonal anti-human antibodies used in this study.**

| Primary antibody | Type | Dilution | Catalogue nr.* |
|---|---|---|---|
| TLR1 | polyclonal rabbit IgG | 1:50 | sc-30000 |
| TLR2 | polyclonal rabbit IgG | 1:50 | sc-8689 |
| TLR2 | polyclonal goat IgG | 1:50 | sc-10739 |
| TLR3 | polyclonal rabbit IgG | 1:50 | sc-10740 |
| TLR4 | polyclonal rabbit IgG | 1:50 | sc-10741 |
| TLR5 | polyclonal rabbit IgG | 1:50 | sc-10742 |
| TLR6 | polyclonal rabbit IgG | 1:50 | sc-30001 |
| TLR7 | polyclonal rabbit IgG | 1:40 | sc-30004 |
| TLR8 | polyclonal rabbit IgG | 1:50 | sc-25467 |
| TLR9 | polyclonal rabbit IgG | 1:40 | sc-25468 |
| TLR10 | polyclonal rabbit IgG | 1:40 | sc-30198 |
| NF-κB | polyclonal rabbit IgG | 1:150 | sc-114 |

Abbreviation: TLR, Toll-like receptors; IgG, immunoglobulin G; NF-κB, nuclear factor-kappaB.

* = Santa Cruz Biotechnology, Santa Cruz, California, USA

## Evaluation of immunostaining

The expressions of TLR1-TLR10 were analysed using a light microscope (Nikon Eclipse 80i). Results were scored semi-quantitatively and photographed using an attached camera (Nikon DS-Fi1). All samples and stainings were analysed and scored by two of the authors (PR and JH) blinded for clinical data. The staining intensity of the endothelium, basal cell layer, the basement membrane zone as well as the deep and superficial thirds of the epithelium were graded in a four-point scale as 0 = no staining, 1 = staining of approximately 1–33% of cells, 2 = staining of 34–66% and 3 = staining of 67–100%. In the OSCC samples the staining intensities were separately assessed in the infiltrative tumor tissue (infiltrative zone) and adjacent normal appearing squamous epithelium. The staining intensity of the endothelium were assessed throughout the OSCC samples.

## Statistical analysis

Data was analyzed by using GraphPad Prism version 5.00 (GraphPad Inc. San Diego, California, USA). The two-tailed Mann Whitney test and Spearman's rho (rS) were used for the analyses of correlations. The Wilcoxon signed-ranks test was used to compare the differences between the different layers of samples. P-values of less than 0.05 were considered statistically significant.

## Result

### Subjects

30 individuals, clinically and histopathologically diagnosed with oral squamous cell carcinoma (OSCC), were enrolled in the study. 12 of them were female (mean age 68.2 years, range 52–85 years) and 18 were male (mean age 61.4 years, range 31–80 years). The mean age of OSCC patients was 64.0 years (range 31–85). The anatomical distribution of the tumors were the tongue (n = 9), maxilla (n = 6), the mandibular gingiva (n = 6), the floor of mouth (n = 5), the alveolar ridge (n = 2), and the palate (n = 2).

30 healthy controls (HC) were enrolled in the study. Out of which 26 were included in the final analysis, as four of the collected tissues samples were deemed inadequate for

histopathological analysis. Out of the selected healthy controls, 17 were female with the mean age of 32.2 years (range 19–56 years) and nine of them were male with the mean age of 26.1 years (range 18±36 years). The mean age of healthy controls was 30.8 years (range 18–56) (Table 1).

## Expression of TLR1-TLR10

The staining intensities in the different layers are presented in Fig 1 and relevant findings are summarized in Table 3. When comparing the staining intensity of TLR1-TLR10 in the superficial and deep epithelial layers, basal layers, basement membrane (BM) zone, as well as in the endothelium, in general significantly stronger staining of several TLRs was observed in HC samples in comparison to normal appearing squamous epithelium in OSCC samples (Fig 2).

**Epithelium.** In the superficial layers of the epithelium the staining intensities of TLR2 ($P = 0.0039$), TLR5 ($P = 0.025$), and TLR8 ($P<0.0001$) were significantly stronger in HC samples compared to superficial OSCC. In the deep layers of the epithelium the staining intensities of TLR2 ($P = 0.0037$) and TLR8 ($P<0.0001$) were significantly stronger in HC samples compared to the same of adjacent normal appearing squamous epithelium in OSCC samples. In the basal layer, the staining intensity of TLR1 was significantly stronger in HC samples compared to adjacent normal appearing squamous epithelium in OSCC samples ($P = 0.03$) (Fig 1).

**Basement membrane zone.** In the basement membrane (BM) zone, the staining intensity of TLR3 ($P = 0.029$), TLR4 ($P<0.0001$), TLR7 ($P<0.0001$), and TLR8 ($P<0.0001$) were clearly / substantially stronger in HC samples than in the normal appearing adjacent tissue in OSCC samples (Figs 1 and 3).

**Endothelium.** In the endothelium, the staining intensity of TLR4 ($P = 0.0005$), TLR7 ($P = 0.001$), and TLR8 ($P = 0.0076$) were significantly stronger in HC samples than in the normal appearing adjacent tissue in OSCC samples (Figs 1 and 3).

**Comparison between infiltrative zone and adjacent normal appearing epithelium in OSCC samples.** In general, in the adjacent normal appearing epithelium the staining intensity of all TLRs decreased from the superficial layers towards the deeper parts of the epithelium. In the infiltrative zone several TLRs showed an increased staining intensity compared to different layers of the adjacent normal appearing epithelium: the difference was significant in TLR1 (basal cell layer; $P = 0.046$), TLR2 (basal cell layer; $P = 0.015$), TLR4 (deep layer and basal cell layer; $P = 0.016$ and $0.0036$, respectively), TLR8 (basal cell layer; $P = 0.0021$) and TLR9 (basal cell layer; $P = 0.035$).

In superficial layer of adjacent normal appearing epithelium in OSCC samples the staining intensity of TLR2 ($P = 0.018$) and TLR7 ($P = 0.016$) were significantly stronger compared to the infiltrative zone. The staining intensities in the different layers in OSCC are presented in Fig 4.

## Association of Candida colonization and the expression of TLR1-10

*Candida* colonization was seen in 23% of the OSCC patients and 4% of the healthy controls. In the normal appearing adjacent tissue in OSCC samples, there was a significant positive correlation between *Candida* colonization and TLR4 expression in the BM zone ($P = 0.012$).

## Expression of NF-κB

The expression of NF-κB was significantly stronger in HC samples compared to the normal appearing adjacent OSCC samples in the superficial and basal layer of the epithelium ($P = 0.02$ and $0.0064$, respectively). The expression of NF-κB was significantly stronger in the infiltrative zone compared to the adjacent normal appearing mucosa in OSCC in all the epithelial layers (superficial, $P = 0.008$; deep, $P = 0.003$; basal, $P = 0.01$) (Table 3).

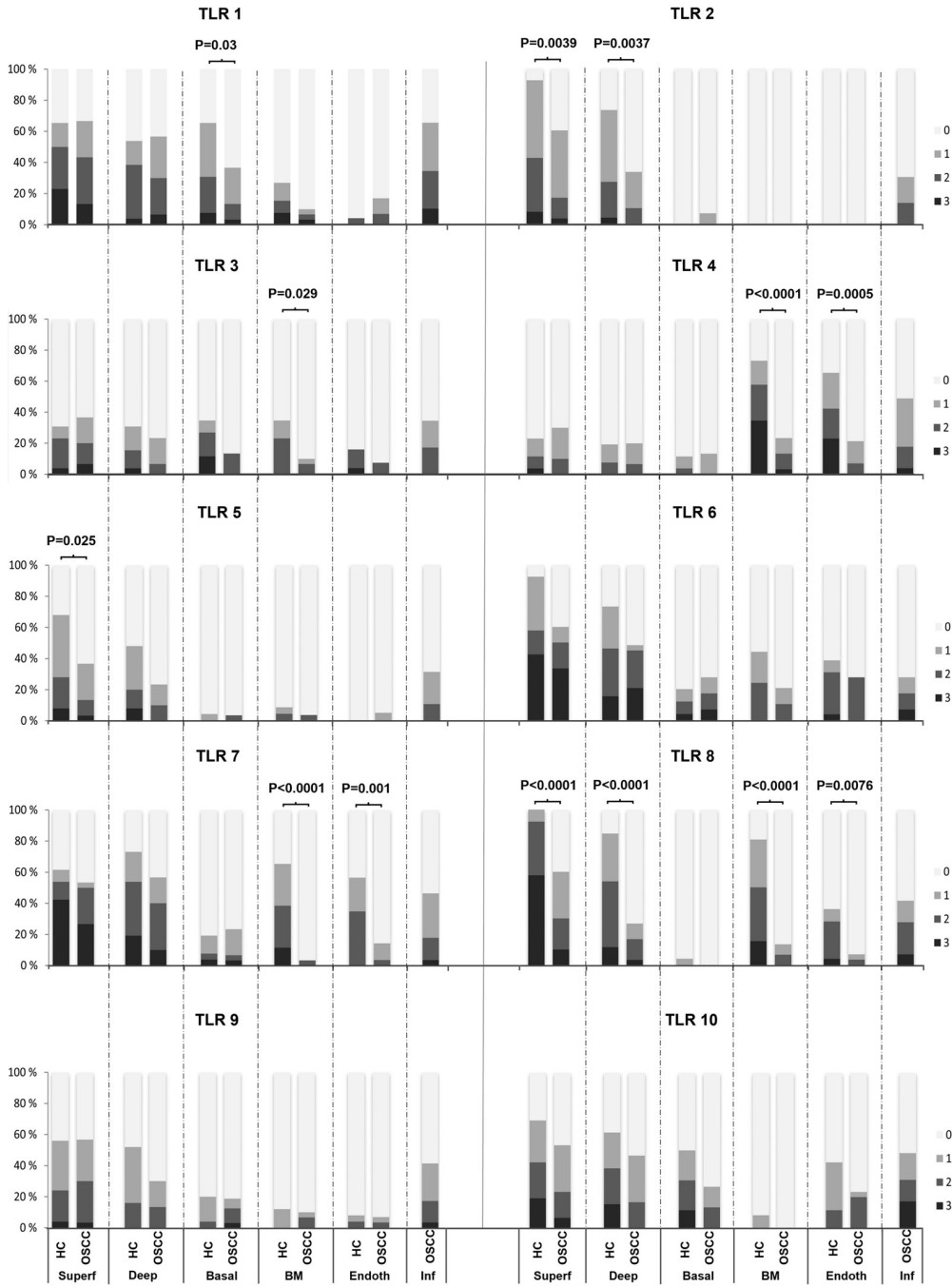

**Fig 1. The staining percentages of TLR1-TLR10 in epithelial layers, basement membrane (BM) zone, endothelium and in the infiltrative zone.** In general, the staining intensity of several TLRs was significantly stronger in healthy controls (HC; n = 26) compared to oral squamous cell carcinoma (OSCC; n = 30) samples. The expression was determined according to the staining intensity in a four-point scale as 0 = no staining, 1 = staining of approximately 1–33% of cells, 2 = staining of 34–66% of cells and 3 = staining of 67–100% of cells. The epithelium was divided into three different layers: the superficial (Superf), deep and basal epithelial layers as well as the basement membrane zone (BM) and endothelium (Endoth). In the OSCC samples the infiltrative zone (Inf) was assessed separately. The P-values in the differences in expression of TLR1-TLR10 in the two patient groups are presented (Mann-Whitney t-test).

**Table 3. The statistically significant differences in the immunostaining of TLR1-10 and NF-κB in healthy control (HC) and oral squamous cell carcinoma (OSCC) samples.**

| Layer | HC compared to adjacent OSCC | | Correlations in adjacent OSCC samples | Infiltrative zone compared to adjacent OSCC | Correlations in infiltrative zone in OSCC samples |
|---|---|---|---|---|---|
| | TLR | NF-κB | | | |
| Superficial epithelium | **TLR2** ↑<br>*P* = 0.0039<br>**TLR5** ↑<br>*P* = 0.025<br>**TLR8** ↑<br>*P*<0.0001 | **NF-κB** ↑<br>*P* = 0.02 | | **NF-κB** ↑<br>*P* = 0.008 | |
| Deep epithelium | **TLR2** ↑<br>*P* = 0.0037<br>**TLR8** ↑<br>*P*<0.0001 | | | **NF-κB** ↑<br>*P* = 0.003 | |
| Basal cell layer | **TLR1** ↑<br>*P* = 0.03 | **NF-κB** ↑<br>*P* = 0.0064 | | **NF-κB** ↑<br>*P* = 0.01 | NF-kB and TLR9<br>NF-kB and TLR10 |
| BM zone | **TLR3** ↑<br>*P* = 0.029<br>**TLR4** ↑<br>*P*<0.0001<br>**TLR7** ↑<br>*P*<0.0001<br>**TLR8** ↑<br>*P*<0.0001 | | *Candida* and TLR4 | | |
| Endothelium | **TLR4** ↑<br>*P* = 0.0005<br>**TLR7** ↑<br>*P* = 0.001<br>**TLR8** ↑<br>*P* = 0.0076 | | | | |

Correlations between TLRs and NF-κB were found only in the infiltrative zone of OSCC samples. In the superficial OSCC samples, there was a significant positive correlation between *Candida* colonization and TLR4 expression in the basement membrane (BM) zone. *P*-values of less than 0.05 were considered statistically significant. Adjacent OSCC: normal appearing adjacent tissue in OSCC sample, ↑: significant stronger staining intensity.

### Associations between expressions of TLR1-10 and NF-κB

There was a positive correlation between the expression of NF-κB and TLR9 as well as TLR10 in the basal layer of the infiltrative zone in OSCC samples (*P* = 0.04 and *P* = 0.002, respectively).

However, no significant correlations were seen in the staining intensities of TLR1-10 and NF-κB in any of the epithelial layers of the HC samples or the adjacent normal appearing mucosa of OSCC samples (Table 3).

## Discussion

The present study analyzed the staining intensity and immunolocalization of TLR1-10 and NF-κB in the different epithelial layers, basement membrane (BM) zone, endothelium, and infiltrative zone of oral squamous cell carcinoma (OSCC) tissue sections and results were compared with the staining intensity in healthy oral mucosa from control patients. In addition, it was analyzed whether the presence of *Candida* colonization affects TLR expression in oral squamous cell carcinoma.

In general, the healthy appearing tissue in OSCC samples showed lower staining intensity of several TLRs compared to the healthy controls (HC). The difference was statistically significant in the staining intensity of TLR1, TLR2, TLR3, TLR4, TLR5, TLR7, and TLR8.

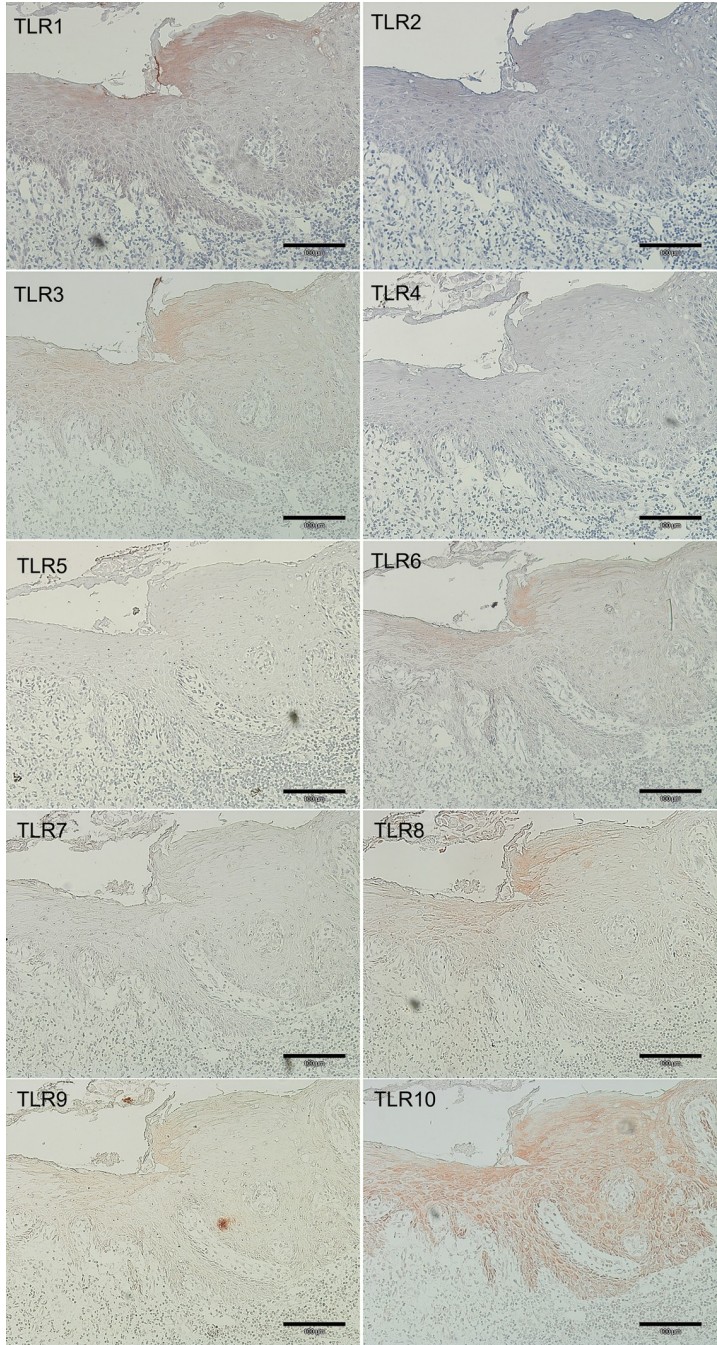

**Fig 2. Staining intensity of TLR1-10 in oral squamous cell carcinoma (OSCC) samples.** The staining intensity of TLR1, TLR2, TLR3, TLR4, TLR5, TLR7 and TLR8 in the normal appearing adjacent tissue in OSCC samples showed significantly lower staining intensity compared to the samples of healthy controls.

The BM zone showed the most variation as the staining intensity of TLR 3, 4, 7, and 8 were significantly decreased in normal appearing adjacent tissue in OSCC samples compared to those of healthy control samples. Different studies have demonstrated the presence of soluble forms of toll-like receptors in saliva, plasma, breast milk, amniotic fluid, and other body fluids

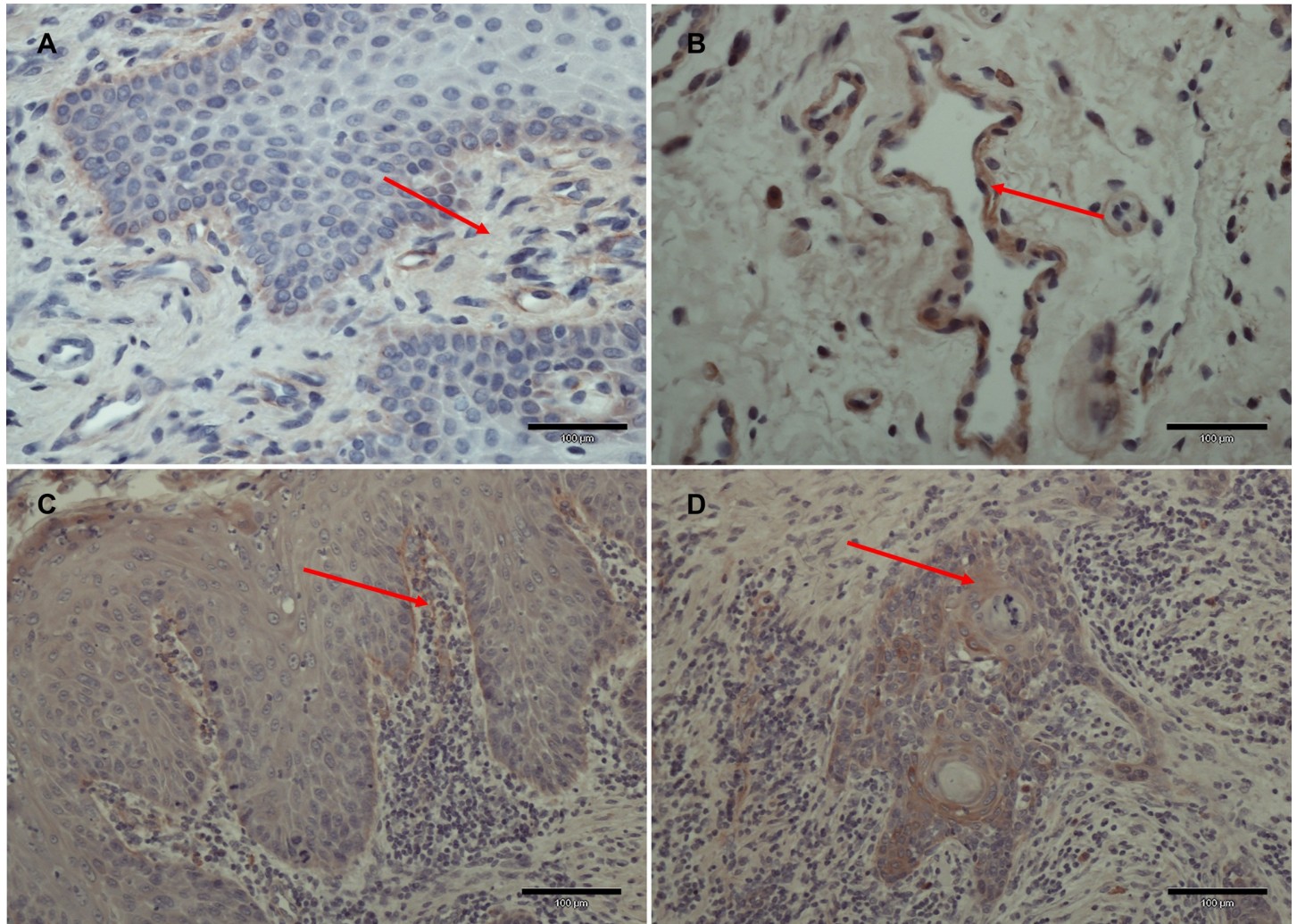

**Fig 3. The staining for TLR4 in healthy control (HC) and oral squamous cell carcinoma (OSCC) samples.** Strong staining intensity of TLR 4 in HC samples in A) BM zone and B) endothelium (red arrow, x40). C) TLR4 positive staining in the BM zone in the healthy appearing tissue in OSCC sample (red arrow, x20). D) TLR 4 positive staining in OSCC infiltrative zone (red arrow, x20). HC: healthy control, OSCC: oral squamous cell carcinoma, BM: basement membrane.

[18, 20]. Therefore, it is possible to speculate that BM zone also harbor soluble forms of TLR fragments produced by the basal epithelial cells and were stained during the analysis.

In the epithelium, we did not find any statistical differences in TLR4 staining intensities between HC and OSCC samples. This is in contrast with the study of Yang et al. 2016, who demonstrated that the expression of cytoplasmic TLR4 is increased in tissue samples of OSCC compared to healthy oral mucosa [41]. The differences might be attributed to the different sampling areas of interest: in the study of Yang et al., the TLR4 expression was scored only from the epithelium. In our study, the TLR4 expression was scored also from the BM-zone, endothelium, and infiltrative zone, separately.

In healthy appearing tissue samples from OSCC patients with candidal colonization at the tumor site, the staining intensity of TLR4 was significantly increased at the BM zone when compared to OSCC samples with no *Candida* ($P = 0.01$). This is in line with the study where epithelial cells of the oral mucosa have been shown to upregulate TLR4 upon stimulation of *Candida albicans* [42]. This might be due to the higher levels of the mutagenic acetaldehyde:

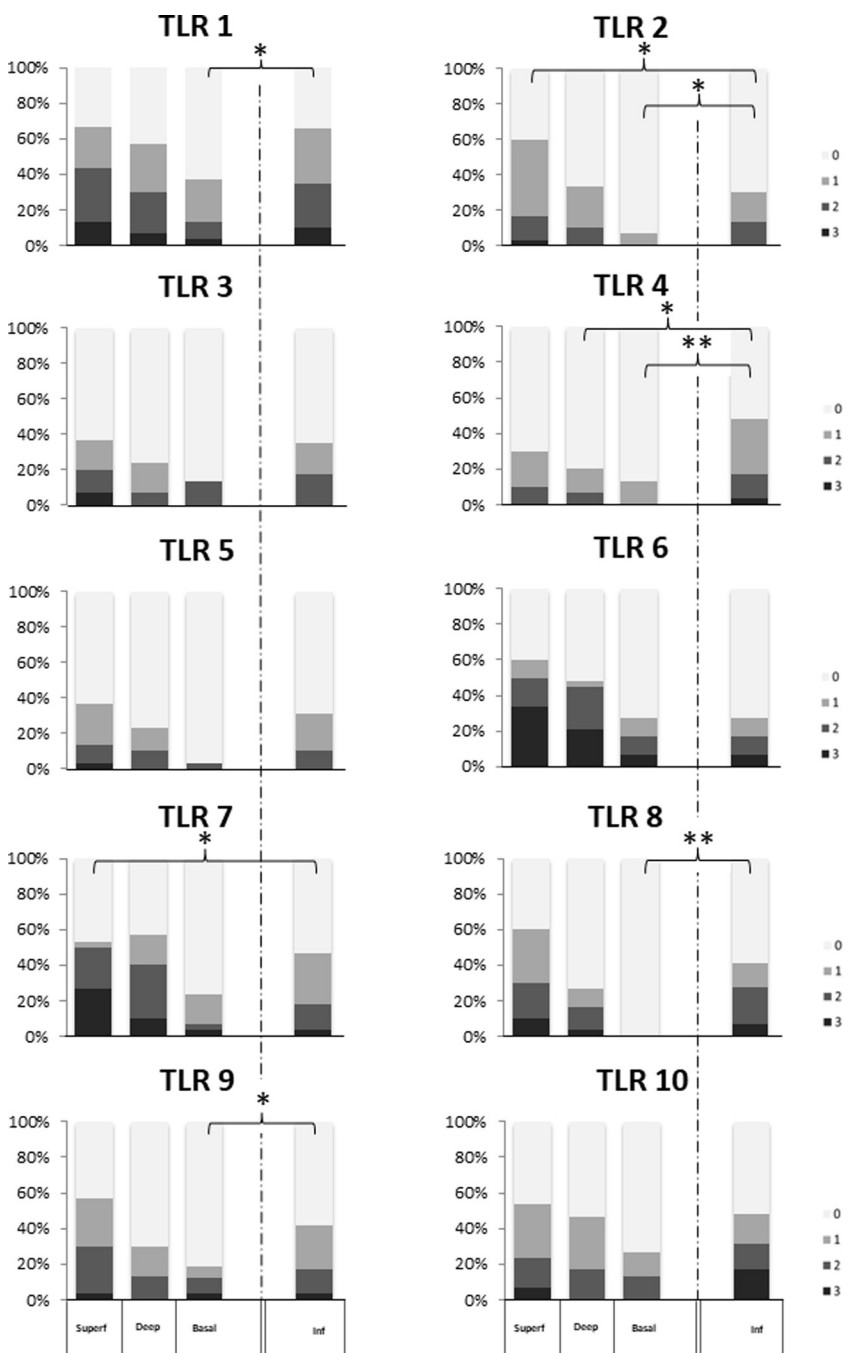

**Fig 4. Comparison of the staining percentages of TLR1-TLR10 between infiltrative zone (inf) and adjacent normal appearing epithelium in oral squamous cell carcinoma samples (OSCC; n = 30).** In the infiltration zone TLR1, TLR2, TLR4, TLR8 and TLR9 showed significantly increased staining intensity compared to the basal cell layer of adjacent normal appearing epithelium. Similarly, the staining intensity of TLR4 was significantly higher in the infiltration zone compared to the deep epithelial layer of adjacent normal appearing epithelium. In opposite, TLR2 and TLR7 showed an increased staining intensity in the superficial layer compared to the infiltration zone. The expression was determined according to the staining intensity in a four-point scale as 0 = no staining, 1 = staining of approximately 1–33% of cells, 2 = staining of 34–66% of cells and 3 = staining of 67–100% of cells. The epithelium was divided into three different layers: the superficial (Superf), deep and basal epithelial layers. The P-values in the differences in expression of TLR1-TLR10 are presented (Mann-Whitney t-test): * = $P < 0.05$; ** = $P < 0.005$.

We have previously demonstrated that a significantly more quantity of mutagenic acetaldehyde is produced by the *Candida* samples collected from OSCC and oral lichenoid lesions sites of the patients than those from the patients with no candida colonization [6].

When comparing infiltration zone with adjacent normal appearing epithelium in OSCC tissue, several TLRs showed an increased staining intensity in the infiltration zone compared to the basal cell layer of adjacent normal appearing epithelium. The difference was statistically significant in TLR1, TLR2, TLR4, TLR8 and TLR9. In addition, the staining intensity of TLR4 was significantly higher in the infiltration zone compared to the deep epithelial layer of adjacent normal appearing epithelium. These findings are in line with the earlier studies which compared the expression of TLRs between tumoral tissue and adjacent normal epithelial tissue in OSCC samples [43–45].

The staining for NF-κB in the superficial epithelium and basal cell layer was significantly lower in the normal appearing adjacent tissue in OSCC samples compared to the control samples ($P = 0.02$, $P = 0.0064$, respectively). This might be attributed to the lower staining for several TLRs as seen in our study. Nevertheless, the expression of NF-κB was significantly stronger in the infiltrative zone compared to the healthy appearing adjacent tissue in OSCC samples in all the epithelial layers. This may indicate that NF-kB contributes to tumor growth and hematologic and lymphatic metastasis [46]. In fact, as seen in our study, the staining intensity of most TLRs in the infiltrative zone was more intense compared to the basal layer of the healthy appearing tissue in OSCC samples. Significant association between NF-κB staining and TLR9 and TLR10 was seen in basal layer in the samples from patients with OSCC. However, no such association was seen in control samples.

When comparing our results to a very limited number of previous studies that used immunohistochemical methods to define the differences in TLR staining intensity and immunolocalization between OSCC and healthy control tissue, we find points that need to be studied in more detail. Several studies have used the tissue adjacent to the tumor as healthy tissue controls [45, 47, 48]. Carcinogenesis is a multifactorial cascade which is affected by numerous intrinsic and extrinsic factors. These factors lead to the loss of control of cell cycle and finally to tumor progression. Many of these factors also affect other parts of the oral mucosa leading to the fact that the normal appearing mucosa adjacent to the tumor is also altered. The results of our previous study supports this notion: staining intensity of tumor suppressor protein p53 was found to be noticeably elevated in both the infiltrative zone and healthy appearing adjacent mucosa of OSCC samples [49]. Tumor suppression protein p53 is a transcription factor involved in apoptosis and control of cell cycle and plays a vital role in the preservation of genetic stability and thus prevention of progression of cancer [50].

We acknowledge some limitations in our study, such as the difference in age between the OSCC patients and healthy controls. This could partly be attributed to the fact that OSCC is more common in the adult population, whereas the control samples taken during the surgical removal of third molars, is mainly performed in young adults. However, it is unlikely that the age gap would have a major impact on the findings because, in contrast to adaptive immune responses, the innate responses are not generally significantly affected by aging [51]. In fact, in this study, we did not find any age-related correlations with the immunostaining of any TLRs. A limitation of this study is that in this study only immunohistochemical staining was used to compare the level of expression of TLR and NF-kB between the OSCC and HC samples. The expression of TLR and NF-kB should ideally also be analyzed using additional techniques, such as measuring mRNA levels by qPCR.

OSCC is a multifactorial disease linked with various risk factors and no single factor has been recognized as having independent influence on the prognosis of OSCC. During the progression of oncogenesis from a normal healthy cell to a pre-malignant or a potentially

malignant cell, mutations in several DNA molecules occur leading to the loss of control of the growth of cell which eventually lead to uncontrolled proliferation of the cells leading to the cancer. To determine the genetic mechanisms involved in OSCC, the mutations should ideally also be analyzed using genetic techniques such as qPCR in order to measure mRNA levels in the tissue sample. In this study only immunohistochemical staining was used to compare the level of expression of TLR and NF-kB between the OSCC and HC samples and is one of the limitations of this study. However, qPCR techniques do not provide information about the specific site within the tissue sample from where the signal is originated. While, in this study we have analyzed immunolocalization and the staining intensity of TLR1-10 and NF-κB in the different epithelial layers, basement membrane (BM) zone, endothelium, and infiltrative zone of tissue sections. In addition, a limitation of our study is the low number of sample size due to the limited amount of tissue available.

In conclusion, all the TLRs and NF-κB, and their co-localization in the epithelium, basement membrane zone and endothelium were mapped for the first time. The differences in the expression of several TLRs and NF-κB between the infiltrative zone and healthy appearing adjacent tissue in OSCC samples forms a significant finding. Additional studies are required to determine the role of soluble forms of TLRs observed in BM zone.

## Supporting information

**S1 File.**
(XLSX)

## Acknowledgments

Authors thank Marjatta Kivekäs for her expert technical support in the laboratory.

## Author Contributions

**Conceptualization:** Peter Rusanen, Emilia Marttila, Johanna Uittamo, Riina Rautemaa-Richardson.

**Data curation:** Peter Rusanen, Emilia Marttila, Jaana Hagström, Johanna Uittamo.

**Funding acquisition:** Justus Reunanen, Riina Rautemaa-Richardson, Tuula Salo.

**Investigation:** Peter Rusanen, Emilia Marttila, Jaana Hagström, Johanna Uittamo.

**Methodology:** Peter Rusanen, Emilia Marttila, Jaana Hagström, Johanna Uittamo, Riina Rautemaa-Richardson.

**Project administration:** Riina Rautemaa-Richardson, Tuula Salo.

**Resources:** Riina Rautemaa-Richardson.

**Supervision:** Justus Reunanen, Tuula Salo.

**Validation:** Jaana Hagström.

**Visualization:** Peter Rusanen, Emilia Marttila.

**Writing – original draft:** Peter Rusanen, Emilia Marttila, Sajeen Bahadur Amatya.

**Writing – review & editing:** Peter Rusanen, Emilia Marttila, Sajeen Bahadur Amatya, Justus Reunanen, Riina Rautemaa-Richardson.

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
