## [Decision Letter · Decision Letter 0]

21 Nov 2022

PONE-D-22-29300Expression Of Toll-like Receptors In Oral Squamous Cell CarcinomaPLOS ONE

Dear Dr. Rusanen,

Thank you for submitting your manuscript to PLOS ONE. After careful consideration, we feel that it has merit but does not fully meet PLOS ONE’s publication criteria as it currently stands. Therefore, we invite you to submit a revised version of the manuscript that addresses the points raised during the review process.

improve your manuscript as per the suggestion of the reviewers.

We look forward to receiving your revised manuscript.

Kind regards,

Aditya K. Panda, Ph.D.

Academic Editor

PLOS ONE

https://helda.helsinki.fi/handle/10138/310852

In your revision ensure you cite all your sources (including your own works), and quote or rephrase any duplicated text outside the methods section. Further consideration is dependent on these concerns being addressed.

“J.R. thanks Academy of Finland for grants 328768 and 299749. RR-R thanks NIHR Manchester Biomedical Research Centre for financial support.”

“This work was supported by the Finnish Dental Society Apollonia (https://www.apollonia.fi/en/grants/) and the Helsinki University Central Hospital (HUCH) Research Funds (https://www.hus.fi/en/research-and-education/research). RR-R was supported by NIHR Manchester Biomedical Reseach Centre. The funders had no role in study design, data collection, and analysis, decision to publish, or preparation of the manuscript.  “

Additional Editor Comments:

Lack of novelty, which has been pointed out by both reviewers.

Reviewers' comments:

Reviewer's Responses to Questions

**Comments to the Author**

1. Is the manuscript technically sound, and do the data support the conclusions?

Reviewer #1: No

Reviewer #2: Yes

2. Has the statistical analysis been performed appropriately and rigorously? 

Reviewer #1: No

Reviewer #2: Yes

3. Have the authors made all data underlying the findings in their manuscript fully available?

Reviewer #1: Yes

Reviewer #2: Yes

4. Is the manuscript presented in an intelligible fashion and written in standard English?

Reviewer #1: No

Reviewer #2: No

5. Review Comments to the Author

Reviewer #1: In this study by Rusanen et al, the authors have determined the expression of TRL1 -10 and NF-κB 38 between OSCC and healthy oral mucosa and correlated this with candida colonization. The data suggests that other than TLR4, other TLRs are downregulated in OSCC tissues, interestingly NF-κB expression is lower in OSCC tissues as compared to normal counterpart. Further, the upregulated TLR4 expression was also correlated with infection candida. Here are the major concerns which severely decrease the importance of the study

• The study is not novel. The expression of TLRs in OSCC tissues are extensively studied which is summarized in review articles (Rich et al., Frontiers in immunology, 2014). In fact some of the studies have observations which is completely different than this study.

• Only 30 OSCC patient and 26 healthy samples were considered in this study. These are not enough to have a conclusion the way it is claimed in this study. What is statistical power in this study?

Reviewer #2: Poor image quality, low resolution of the images and font quality needs enhanced. The original figs are fine, however in the pdf they are not good.

How do this study related to existing literature.

Role of TLR is extensively studied, how this study is different?

How the expressions are related to databases (TCGA or Oncomine)?

6. PLOS authors have the option to publish the peer review history of their article (what does this mean?). If published, this will include your full peer review and any attached files.

Reviewer #1: No

Reviewer #2: No

---

## [Author Response · Author response to Decision Letter 0]

6 Aug 2023

Dear Editor,

The authors thank you very much for considering this manuscript titled ‘Expression of Toll-like Receptors in Oral Squamous Cell Carcinoma’.

The manuscript has now been revised according to suggestions and comments from the reviewers.

We hope the article is now ready for publication.

Editor comment 1

Author response:

PLOS ONE’s style requirements have been considered while preparing the manuscript.

Editor comment 2

We noticed you have some minor occurrence of overlapping text with the following previous publication(s), which needs to be addressed:

Author response:

The overlapping text might be due to our previous study that has exactly the same study design as this one, with the difference that now we compare the TLRs staining intensity with OSCC instead of OLD. 

Editor comment 3

We note that the grant information you provided in the ‘Funding Information’ and ‘Financial Disclosure’ sections do not match.

Author response:

The grant information has been corrected.

Editor comment 4

Please include your full ethics statement in the ‘Methods’ section of your manuscript file. In your statement, please include the full name of the IRB or ethics committee who approved or waived your study, as well as whether or not you obtained informed written or verbal consent. If consent was waived for your study, please include this information in your statement as well.

Author response:

Ethical approval number has been added.

Editor comment 4

Please remove any funding-related text from the manuscript and let us know how you would like to update your Funding Statement.

Author response:

Funding related text has been corrected.

In acknowledgements a line has been added as:

Authors thank Marjatta Kivekäs for her expert technical support in the laboratory.

Reviewer #1

Authors thank reviewer 1 very much for his time and effort in reviewing the manuscript. Authors would also like to thank him for all his valuable suggestions. The comments provided have been addressed point by point and the suggestions have been implemented.

Reviewer #1 comment 1

In this study by Rusanen et al, the authors have determined the expression of TRL1 -10 and NF-κB 38 between OSCC and healthy oral mucosa and correlated this with candida colonization. The data suggests that other than TLR4, other TLRs are downregulated in OSCC tissues, interestingly NF-κB expression is lower in OSCC tissues as compared to normal counterpart. Further, the upregulated TLR4 expression was also correlated with infection candida. Here are the major concerns which severely decrease the importance of the study

• The study is not novel. The expression of TLRs in OSCC tissues are extensively studied which is summarized in review articles (Rich et al., Frontiers in immunology, 2014). In fact some of the studies have observations which is completely different than this study.

Author response:

Authors agree that the expression of TLRs in OSCC tissue are extensively studied, but ours is the first study where TLR1-10 have been compared between OSCC infiltration, healthy appearing mucosa in OSCC and healthy controls using immunohistochemical staining. In addition to epithelium, basement membrane zone and endothelium have also been studied. Our study mapped TLR1-10, and also the soluble forms of TLRs in the basement membrane zone in OSCC and HC.

In the study performed by Yang et al. 2016 and Visioli et al. 2022 [41 and 44, respectively] the TLR4 expression was significantly higher in the OSCC compared to the healthy control tissue that had taken during the extraction of the impacted wisdom tooth. In our study we found no significant differences in TLR expression between these layers in the epithelium. 

We have run whole new comparison between the infiltration and adjacent normal appearing epithelium in OSCC tissue. Figure number 4 have been added to the manuscript to support our findings. These findings are in line with the earlier studies which compared the expression of TLRs between tumoral tissue and adjacent normal epithelial tissue in OSCC samples [43–45]. 

Our results are not in opposition to the previous studies. The misinterpretation might have appeared due to the differences in TLR staining intensity between OSCC and HC in the basement membrane zone and endothelium. The authors did not find studies that support or oppose our findings in the basement membrane zone and endothelium. 

In the manuscript the changes have been made as (in page number 11 and line number 306):

Comparison between infiltrative zone and adjacent normal appearing epithelium in OSCC samples

In general, in the adjacent normal appearing epithelium the staining intensity of all TLRs decreased from the superficial layers towards the deeper parts of the epithelium. In the infiltrative zone several TLRs showed an increased staining intensity compared to different layers of the adjacent normal appearing epithelium: the difference was significant in TLR1 (basal cell layer; P=0.046), TLR2 (basal cell layer; P=0.015), TLR4 (deep layer and basal cell layer; P=0.016 and 0.0036, respectively), TLR8 (basal cell layer; P=0.0021) and TLR9 (basal cell layer; P=0.035).

In superficial layer of adjacent normal appearing epithelium in OSCC samples the staining intensity of TLR2 (P=0.018) and TLR7 (P=0.016) were significantly stronger compared to the infiltrative zone. The staining in-tensities in the different layers in OSCC are presented in Fig 4.

Figure 4. Comparison of the staining percentages of TLR1-TLR10 between infiltrative zone (inf) and adjacent normal appearing epithelium in oral squamous cell carcinoma samples (OSCC; n=30). In the infiltration zone TLR1, TLR2, TLR4, TLR8 and TLR9 showed significantly increased staining intensity compared to the basal cell layer of adjacent normal appearing epithelium. Similarly, the staining intensity of TLR4 was significantly higher in the infiltration zone compared to the deep epithelial layer of adjacent normal appearing epithelium. In opposite, TLR2 and TLR7 showed an increased staining intensity in the superficial layer compared to the infiltration zone.

The expression was determined according to the staining intensity in a four-point scale as 0 = no staining, 1 = staining of approximately 1-33 % of cells, 2 = staining of 34-66 % of cells and 3 = staining of 67-100 % of cells. The epithelium was divided into three different layers: the superficial (Superf), deep and basal epithelial layers. The P-values in the differences in expression of TLR1-TLR10 are presented (Mann-Whitney t-test): * = P < 0.05; ** = P < 0.005.

In the manuscript the changes has been made as (in page number 13 and line number 394):

When comparing infiltration zone with adjacent normal appearing epithelium in OSCC tissue, several TLRs showed an increased staining in-tensity in the infiltration zone compared to the basal cell layer of adjacent normal appearing epithelium. The difference was statistically significant in TLR1, TLR2, TLR4, TLR8 and TLR9. In addition, the staining intensity of TLR4 was significantly higher in the infiltration zone compared to the deep epithelial layer of adjacent normal appearing epithelium. These findings are in line with the earlier studies which compared the expression of TLRs between tumoral tissue and adjacent normal epithelial tissue in OSCC samples [43–45]. 

Reviewer #1 comment 2

Only 30 OSCC patient and 26 healthy samples were considered in this study. These are not enough to have a conclusion the way it is claimed in this study. What is statistical power in this study?

Author response:

The sample size could not be achieved in higher number due to limited amount of tissue available. Our findings are in line with the previous studies with similar kind of sample size.

Reviewer #2

Authors thank reviewer 2 very much for his time and effort in reviewing the manuscript. The authors would also like to thank him for all his valuable suggestions. The comments provided have been addressed point by point and the suggestions have been implemented.

Reviewer #2 comment 1

Poor image quality, low resolution of the images and font quality needs enhanced. The original figs are fine, however in the pdf they are not good. 

Author response:

All images and figures have now been uploaded as the requirement of PLOS one guidelines.

 

Reviewer #2 comment 2

How do this study related to existing literature.

Author response

Author response for this comment has been answered in the next comment (comment number 3).

Reviewer #2 comment 3

Role of TLR is extensively studied, how this study is different? 

Author response

Authors agree that the expression of TLRs in OSCC tissue are extensively studied, but ours is the first study where TLR1-10 have been compared between OSCC infiltration, healthy appearing mucosa in OSCC and healthy controls using immunohistochemical staining. In addition to epithelium, basement membrane zone and endothelium have also been studied. Our study mapped TLR1-10, and also the soluble forms of TLRs in the basement membrane zone in OSCC and HC.

We have run whole new comparison between the infiltration and adjacent normal appearing epithelium in OSCC tissue. Figure number 4 have been added to the manuscript to support our findings. These findings are in line with the earlier studies which compared the expression of TLRs between tumoral tissue and adjacent normal epithelial tissue in OSCC samples [43–45]. 

In the study performed by Yang et al. 2016 and Visioli et al. 2022 [41 and 44, respectively] the TLR4 expression was significantly higher in the OSCC compared to the healthy control tissue that had taken during the extraction of the impacted wisdom tooth. In our study we found no significant differences in TLR expression between these layers in the epithelium. However, our findings are not in opposition to the previous studies. 

In the manuscript the changes have been made as (in page number 11 and line number 306):

Comparison between infiltrative zone and adjacent normal appearing epithelium in OSCC samples

In general, in the adjacent normal appearing epithelium the staining intensity of all TLRs decreased from the superficial layers towards the deeper parts of the epithelium. In the infiltrative zone several TLRs showed an increased staining intensity compared to different layers of the adjacent normal appearing epithelium: the difference was significant in TLR1 (basal cell layer; P=0.046), TLR2 (basal cell layer; P=0.015), TLR4 (deep layer and basal cell layer; P=0.016 and 0.0036, respectively), TLR8 (basal cell layer; P=0.0021) and TLR9 (basal cell layer; P=0.035).

In superficial layer of adjacent normal appearing epithelium in OSCC samples the staining intensity of TLR2 (P=0.018) and TLR7 (P=0.016) were significantly stronger compared to the infiltrative zone. The staining in-tensities in the different layers in OSCC are presented in Fig 4.

Figure 4. Comparison of the staining percentages of TLR1-TLR10 between infiltrative zone (inf) and adjacent normal appearing epithelium in oral squamous cell carcinoma samples (OSCC; n=30). In the infiltration zone TLR1, TLR2, TLR4, TLR8 and TLR9 showed significantly increased staining intensity compared to the basal cell layer of adjacent normal appearing epithelium. Similarly, the staining intensity of TLR4 was significantly higher in the infiltration zone compared to the deep epithelial layer of adjacent normal appearing epithelium. In opposite, TLR2 and TLR7 showed an increased staining intensity in the superficial layer compared to the infiltration zone.

The expression was determined according to the staining intensity in a four-point scale as 0 = no staining, 1 = staining of approximately 1-33 % of cells, 2 = staining of 34-66 % of cells and 3 = staining of 67-100 % of cells. The epithelium was divided into three different layers: the superficial (Superf), deep and basal epithelial layers. The P-values in the differences in expression of TLR1-TLR10 are presented (Mann-Whitney t-test): * = P < 0.05; ** = P < 0.005.

In the manuscript the changes have been made as (in page number 13 And line number 394):

When comparing infiltration zone with adjacent normal appearing epithelium in OSCC tissue, several TLRs showed an increased staining in-tensity in the infiltration zone compared to the basal cell layer of adjacent normal appearing epithelium. The difference was statistically significant in TLR1, TLR2, TLR4, TLR8 and TLR9. In addition, the staining intensity of TLR4 was significantly higher in the infiltration zone compared to the deep epithelial layer of adjacent normal appearing epithelium. These findings are in line with the earlier studies which compared the expression of TLRs between tumoral tissue and adjacent normal epithelial tissue in OSCC samples [43–45].

Author response:

Reviewer #2 comment 4

How the expressions are related to databases (TCGA or Oncomine)?

Author response:

The aim of the study was to map all TLRs using IHC-staining in OSCC compared to HC. In the databases the methods are based on PCR. That’s why the authors think that the comparison of TCGA or Oncomine database is not required in this study. 

In the manuscript the changes have been made as (in page number 14 and line number 442):

A limitation of this study is that in this study only immunohistochemical staining was used to compare the level of expression of TLR and NF-kB between the OSCC and HC samples. The expression of TLR and NF-kB should ideally also be analysed using additional techniques, such as measuring mRNA levels by qPCR.

---

## [Decision Letter · Decision Letter 1]

27 Sep 2023

PONE-D-22-29300R1Expression Of Toll-like Receptors In Oral Squamous Cell CarcinomaPLOS ONE

Dear Dr. Rusanen,

Thank you for submitting your manuscript to PLOS ONE. After careful consideration, we feel that it has merit but does not fully meet PLOS ONE’s publication criteria as it currently stands. Therefore, we invite you to submit a revised version of the manuscript that addresses the points raised during the review process.

As suggested by the reviewers the authors need to consider more samples to justify their findings. ==============================

We look forward to receiving your revised manuscript.

Kind regards,

Aditya K. Panda, Ph.D.

Academic Editor

PLOS ONE

Journal Requirements:

Additional Editor Comments:

As suggested by the reviewers authors must enrolled more number samples to justify their findings.

Reviewers' comments:

Reviewer's Responses to Questions

**Comments to the Author**

1. If the authors have adequately addressed your comments raised in a previous round of review and you feel that this manuscript is now acceptable for publication, you may indicate that here to bypass the “Comments to the Author” section, enter your conflict of interest statement in the “Confidential to Editor” section, and submit your "Accept" recommendation.

Reviewer #1: (No Response)

Reviewer #2: (No Response)

2. Is the manuscript technically sound, and do the data support the conclusions?

Reviewer #1: No

Reviewer #2: Partly

3. Has the statistical analysis been performed appropriately and rigorously? 

Reviewer #1: No

Reviewer #2: I Don't Know

4. Have the authors made all data underlying the findings in their manuscript fully available?

Reviewer #1: Yes

Reviewer #2: Yes

5. Is the manuscript presented in an intelligible fashion and written in standard English?

Reviewer #1: No

Reviewer #2: Yes

6. Review Comments to the Author

Reviewer #1: With this small number of sample size it is difficult to have a conclusion and statistical power. So authors have to increase number of samples to achieve logical conclusion.

Reviewer #2: Authors have answered most of the comments without further experimentation. Still the number of samples is a concern.

7. PLOS authors have the option to publish the peer review history of their article (what does this mean?). If published, this will include your full peer review and any attached files.

Reviewer #1: No

Reviewer #2: No

---

## [Author Response · Author response to Decision Letter 1]

16 Jan 2024

Dear Editor,

The authors thank you very much for considering this manuscript titled ‘Expression of Toll-like Receptors In Oral Squamous Cell Carcinoma’.

The manuscript has now been revised according to suggestions and comments from the reviewers.

Kind regards, 

Peter Rusanen

---

## [Editor Report · Decision Letter 2]

28 Feb 2024

Expression Of Toll-like Receptors In Oral Squamous Cell Carcinoma

PONE-D-22-29300R2

Dear Dr. Rusanen,

We’re pleased to inform you that your manuscript has been judged scientifically suitable for publication and will be formally accepted for publication once it meets all outstanding technical requirements.

Kind regards,

Aditya K. Panda, Ph.D.

Academic Editor

PLOS ONE
---

## [Editor Report · Acceptance letter]

29 Mar 2024

PONE-D-22-29300R2 

PLOS ONE

Dear Dr. Rusanen, 

I'm pleased to inform you that your manuscript has been deemed suitable for publication in PLOS ONE. Congratulations! Your manuscript is now being handed over to our production team.

Kind regards, 

on behalf of

Dr. Aditya K. Panda 

Academic Editor

PLOS ONE